# Perspectives of FTIR as Promising Tool for Pathogen Diagnosis, Sanitary and Welfare Monitoring in Animal Experimentation Models: A Review Based on Pertinent Literature

**DOI:** 10.3390/microorganisms12040833

**Published:** 2024-04-20

**Authors:** Matheus Morais Neves, Renan Faria Guerra, Isabela Lemos de Lima, Thomas Santos Arrais, Marco Guevara-Vega, Flávia Batista Ferreira, Rafael Borges Rosa, Mylla Spirandelli Vieira, Belchiolina Beatriz Fonseca, Robinson Sabino da Silva, Murilo Vieira da Silva

**Affiliations:** 1Biotechnology in Experimental Models Laboratory—LABME, Federal University of Uberlândia, Uberlândia 38405-330, MG, Brazil; matheusmoraisneves@gmail.com (M.M.N.); renanfg@ufu.br (R.F.G.); isabela.lemos@ufu.br (I.L.d.L.); thomas.arrais@ufu.br (T.S.A.); flaviabatistaf@yahoo.com.br (F.B.F.); 2Rodents Animal Facilities Complex, Federal University of Uberlandia, Uberlândia 38400-902, MG, Brazil; rafaeluellrosa@gmail.com; 3Innovation Center in Salivary Diagnostic and Nanotheranostics, Department of Physiology, Institute of Biomedical Sciences, Federal University of Uberlandia, Uberlândia 38408-100, MG, Brazil; marco.guevara.vega@gmail.com (M.G.-V.); robinsonsabino@gmail.com (R.S.d.S.); 4Faculty of Medicine, Maria Ranulfa Institute, Av. Vasconselos Costa 321, Uberlândia 38400-448, MG, Brazil; myllaspirandelli@hotmail.com; 5School of Veterinary Medicine, Federal University of Uberlandia, Uberlândia 30402-018, MG, Brazil; biafonseca@ufu.br

**Keywords:** Fourier transform infrared (FTIR), animal facility, sanitary control, welfare monitoring, microorganisms diagnostic

## Abstract

Currently, there is a wide application in the literature of the use of the Fourier Transform Infrared Spectroscopy (FTIR) technique. This basic tool has also proven to be efficient for detecting molecules associated with hosts and pathogens in infections, as well as other molecules present in humans and animals’ biological samples. However, there is a crisis in science data reproducibility. This crisis can also be observed in data from experimental animal models (EAMs). When it comes to rodents, a major challenge is to carry out sanitary monitoring, which is currently expensive and requires a large volume of biological samples, generating ethical, legal, and psychological conflicts for professionals and researchers. We carried out a survey of data from the relevant literature on the use of this technique in different diagnostic protocols and combined the data with the aim of presenting the technique as a promising tool for use in EAM. Since FTIR can detect molecules associated with different diseases and has advantages such as the low volume of samples required, low cost, sustainability, and provides diagnostic tests with high specificity and sensitivity, we believe that the technique is highly promising for the sanitary and stress and the detection of molecules of interest of infectious or non-infectious origin.

## 1. Introduction

Fourier Transform Infrared (FTIR) is a spectroscopy technique with diverse applications; it uses light phenomena, such as adoption, reflection, and emission to study the sample composition [1]. The unique pattern of infrared absorption by a given molecule or functional group produces characteristic bands in its FTIR spectra, which is why it is used to characterize the molecular structures of the target material under study [2,3,4].

The “fingerprint” of the studied molecule is provided by the biochemical profile obtained when it is irradiated by infrared light, from which it absorbs a certain amount of incident radiation at a specific energy/frequency and undergoes vibrational excitation from the ground state to a higher energy state, also known as a higher vibrational state. The profile is drawn according to the vibration mass and can vary according to the type of molecular bond, the intra- and intermolecular environment, and the coupling with other vibrations [4].

The height of the peaks is proportional to the concentration of the corresponding chemical moieties, and the width estimates intermolecular interactions. This technique can analyze proteins, nucleic acids, lipids, and carbohydrates in a biological sample, proving to be very sensitive for detecting small changes in the molecular structure, such as the secondary structure of proteins, the mutation of nucleic acids, and the peroxidation of phospholipids [4,5,6]. This tool can be combined with animal experimentation as it plays a crucial role in the development of science, to act by monitoring the well-being and health of animals, which are parameters that directly impact the quality, reproducibility, and reliability of the data generated [7,8].

Experimental animal models play a crucial role in the development of science and have significant importance in advancing scientific knowledge and understanding various diseases and their treatments [9]. Animal models serve as an essential bridge between basic research and clinical applications. They help researchers validate findings from in vitro studies and test hypotheses in living organisms [10]. Data obtained from animal models can guide the design and implementation of human clinical trials, improving the chances of success and reducing risks associated with human experimentation [11].

However, the use of animals in research must always be associated with the ethical and legal aspects of animal experimentation [12]. To reduce the number of animals, sanitary quality, genetics, and well-being must be premises for user researchers and for all professionals involved in research, from animal husbandry to euthanasia [12,13]. The breeding of laboratory animals requires operating conditions that guarantee the welfare of the animals and their health, following the recommendations of the competent agencies [7,8]. This care is also in line with the principle of the 3Rs [7,14].

To ensure the quality of the animals used in the research, it is necessary to detect infections at the beginning and maintain a routine for this monitoring [15]. Stress monitoring should also follow the same premise, not just focusing on behavioral observations or signs of pain but also identifying markers such as corticosterone [16,17,18]. Recurrent and systematic tests are able to better predict the impacts of these parameters on research [15]. Establishing a monitoring routine seems easy, but it comes up against many details that must be planned, such as facilities that house more than one species, differences in procedures for sampling animals for testing, types of agents to be tested, and type of sample and detection method, which varies according to each microorganism. Collection can be invasive, the method can require a large volume, and the number of samples needed to detect a given infectious agent can vary. Another bias is the cost of techniques such as ELISA and PCR [19].

Faced with the difficulties presented for monitoring animals, FTIR presents itself as an alternative because it is easy to handle, uses a small volume of samples (non-invasive), and guarantees time savings. These characteristics are in line with well-being and refinement, bringing an alternative to establishing a routine for maintaining the health of animals and introducing stress monitoring [1,20].

In this context, the paramount importance of developing new tools for diagnosis and monitoring in the field of laboratory animal science is well recognized. In this review, we synthesize information from various studies pertinent to FTIR that render this methodology promising and impactful in the discussed context. Accordingly, our review aims primarily to present the comparison and advantages of using this tool over those commonly employed in contemporary research. Thus, we focus on the applicability related to sanitary and welfare monitoring provided by such a methodology.

## 2. Fourier-Transform Infrared Spectroscopy (FTIR) Technique

Fourier Transform Infrared (FT-IR) is a spectroscopy technique because it uses light phenomena, such as adoption, reflection, and emission, and your interaction with the matter, to study the sample composition. The most-used infrared radiation frequencies, namely, mid-infrared, range from 4000 cm^−1^ to 400 cm^−1^ (respectively, 2.5 to 25 µm wavelength) with energy rates around 8–40 kJ mol^−1^. This energy is sufficient to promote changes in the vibrational energy levels resulting in molecular vibrations in molecules and molecular ions. Other regions of infrared radiation frequencies are known as far-infrared (<400 cm^−1^) and near-infrared (13,000 cm^−1^ to 4000 cm^−1^). However, only a unique spectrum for each molecule, compound, and/or complex sample as their “fingerprint” can be generated [1].

Therefore, these characteristic spectra (a plot of absorption intensity of the light that passed through the sample versus radiation frequency) are determined by the stretching (changes in the bonds lengths, e.g., symmetric and asymmetric) and bending (changes in spatial bonds angles, e.g., scissoring, rocking, wagging, and twisting) modes of vibrational motion [21]. In this way, the specific vibrations are influenced by these three factors: molecular structure of the sample, chemical environment, and absorptivity [21].

Infrared spectroscopy was introduced in the 1600s as an analytical method for its most common application: qualitative identification of organic compounds by chemists [22]. The progress of computer technology has enabled the association of modern spectrophotometers, and the mathematical method introduced by J.B.J. Fourier in the 1880s has contributed to the development and effectiveness of infrared spectroscopy overcoming barriers for its practical application [22].

However, the first Fourier spectrum was generated only in the late 1940s using the Michelson interferometer setup, and over the next twenty years, the first low-cost generation of Fourier-transform infrared spectrometers was developed, growing commercial interest [23]. The use of nanogram levels (microsamples) and the speed to provide infrared spectra are the principal advantages of FT-IR over dispersive spectrometers. So, those non-time-consuming devices can generate a spectrum in less than 1.0 s, making it possible to obtain a number of spectra for the same sample, improving the signal-to-noise ratio [1].

Fourier-transform infrared spectrometers operate differently from others in terms of the design of the optical pathway, which includes an interferometer (Figure 1A). In this kind of device, in the interferometer, the beam splitter splits the infrared beam reflected by two mirrors (fixed and moved) at the same time. Therefore, both beams recombine constructively or destructively, and pass through the sample that receives all possible radiation frequencies simultaneously. The result is a plot of intensity versus time, called an interferogram. However, this temporal domain signal is not human readable, being necessarily a computational process, which involves Fourier Transform mathematical principles, responsible for separating the individual absorption frequencies from the generated plot. Thus, a spectrum of intensity versus frequency is obtained [1].

The FTIR technique has advantages and limitations that motivate their use for specific applications. According to Fadlelmoula and Pinho [24] (2021), the first application of FTIR as a potential method dates from 1972, and the diagnosis application in the biological field dates from 10 years later. In general, the most common biological samples studied are blood, serum, plasma, urine, and tissues, i.e., liquid fluids and solid material. The use of IR radiation to interact with this kind of sample characterizes the principal weakness of this technique: limited surface sensitivity and reproducibility of the spacer thickness [25]. For aqueous samples, it is limited to 6.0 µm (micrometer scale), especially for the high absorbance of water molecules (ε = 104.400 cm^2^ mol^−1^) [25,26].

To overcome this limitation, a complementary sampling technique for FTIR, named Attenuated Total Reflectance (ATR), was developed [27]. Despite the conventional FTIR technique, which works without an incidence angle between the IR source and the sample (transmission operation mode), in the ATR-FTIR method, the IR light is not reflected directly on the sample surface (reflection operation mode) (Figure 1B). The IR radiation is reflected by an internal reflection element (IRE) with the following material requirements: transparent for IR radiation range, high refractive index, and excellent transmitting properties [28]. The most common ATR crystal materials are Germanium, Thallium Bromo-Iodide (KRS-5), Zinc Selenide (ZnSe), Diamond, and Silicon. Furthermore, sample preparation also can be required to eliminate water molecules using physical procedures such as heating and drying.

**Figure 1 microorganisms-12-00833-f001:**
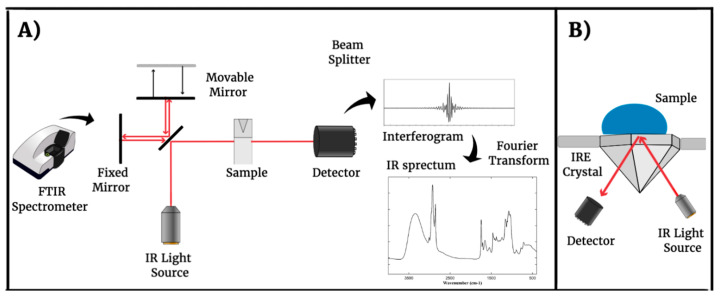
(**A**) Double Mirror Michelson Interferometer as applied in FTIR analysis. (**B**) Principle of Internal Reflection mode in IR spectroscopy [28].

Besides the limitations, the ATR-FTIR has become an established spectroscopy technique for bioanalytical applications because of its strengths [24]. It is related primarily to the high chemical specificity of the samples improving the sensitivity of the IR spectra obtained [29]. Additionally, in terms of the practical aspects of instrumentation and analysis, the experimental procedures are characterized by simplicity, non-invasive, and low-cost time demands, and the spectrophotometers are characterized by mechanical simplicity and calibration facility. Thus, ATR-FTIR measurements make the analysis much more accurate and reproducible [20]. In this way, the FT-IR has become an invaluable tool and an important analytical technique, with reliable and promising applications in several fields of study, as well as those described in the next Section.

## 3. Infections by Different Classes of Pathogens Can Be Diagnosed Using FTIR on Biological Samples

The first studies with FTIR and microbiology date back to the 1950s, but it was in the 1990s that Naumann and collaborators established the experimental conditions for preparing samples and analyzing data for microbiological applications [30,31]. Likewise, the need to identify clinical microbiology agents dates back years, but even today it is necessary to elucidate precise and reliable methods. Studies on FT-IR developments for bacterial discrimination are mostly based on the identification of bacterial groups previously defined by phenotypic or genotypic typing methods [31].

The distinctiveness of this tool can be exemplified in the work of Rebuffo et al. [32] in distinguishing different species of the Listeria genus, and Nitrosetein et al. [33] in distinguishing two important species of enterococci pathogens. This ability allows normal microbiota to be separated from important pathogenic and opportunistic microorganisms in one sample, providing an efficient and reliable diagnosis when necessary.

Another example was research carried out to discriminate strains of *Staphylococcus aureus* and coagulase-negative Staphylococci and other pathogens by FTIR, where they sought a robust method and managed to build an identification model that provided 100% sensitivity and 98% specificity, being able to identify the pathogen among other taxonomically close species, species of the same genus, and taxonomically distant species of different genera. This study obtained 188 spectra of the target pathogen (*S. aureus*) and 263 spectra of other Staphylococci, with protein/amide peaks I e II (1700–1500 cm^−1^), phospholipids/DNA/RNA (1500–1185 cm^−1^), and polysaccharides (1185–900 cm^−1^) [34].

In addition to the possibility of diagnosing and distinguishing common pathogens, FTIR can also be used as a monitoring methodology for the emergence of possible mutations causing epidemics, thus controlling them at their outset. This is feasible for both retrospective studies and real-time surveillance, as shown by Guerrero-Lozano et al. [35] and Lombardo et al. [36], respectively, for *Acinetobacter baumannii*.

Another study with bacteria discussed the preparation of microbial samples, which and these organisms take a long time to grow on plates or culture bottles. In this work, AlRabiah and collaborators raised the importance of identifying species of bacteria common in human infections, such as those related to the urinary tract. They reported a microculture technique that, in addition to solving the hassle of sample preparation, also achieved good results in obtaining the metabolic profiles of *Escherichia coli*, with high reproducibility. Through FTIR, they were also able to discriminate the species of this bacteria, mainly those involved in urinary tract infections [37].

Passaris and collaborators also achieved good results for the capsular typing of *Streptococcus pneumoniae*, which is an opportunistic, Gram-positive pathogen that causes a wide variety of diseases, from sinusitis, otitis media, and conjunctivitis, to more serious diseases such as pneumonia, bacteremia, and meningitis [38,39]. A 98.0% accuracy was achieved in determining the serotypes represented in the training set, which consisted of 34 different serotypes and included all 24 vaccine types and 10 non-vaccine types. Species outside the training set were also used, and the technique was able to distinguish the bacteria of interest from this other set. These results also reinforce the potential of the technique for this application [38].

A molecular signature has also been obtained for bacteria of the genus Bacillus, which are pathogenic, Bacillus anthracis and Bacillus cereus. These two species are related and present difficulties in distinguishing them using conventional methods, making it necessary to search for methods for this type of identification due to the pathogenic potential of these species. A pioneering study managed to determine well-separated groups of the two species using FTIR. The region of the FTIR spectrum that showed the discrimination of *B. anthracis*/*B. cereus* isolates was found to be in the region 1300–700 cm^−1^, corresponding to the absorption wave numbers of polysaccharides [40]. Once again, the technique is capable of discriminating pathogens of interest to health; in this case, it was applied to humans but can be extrapolated to animals.

The FTIR technique has also been very well used for the detection of bacteria belonging to the *Mollicutes* class (such as *Mycoplasma*). Furthermore, within the same species, the technique was used with precision for fine discrimination of strains, with peaks of saccharide 1200–900 cm^−1^ [41].

Filamentous fungi and yeasts can also now be studied based on FTIR, showing potential for use as a modern technique for rapid identification and characterization of these in hospitals, health centers, clinical analysis laboratories, and the food, feed, beverage, and water industries, being an effective tool for rapid identification and quantification of biocompounds produced by fungi [42]. A study carried out by Tralamazza and collaborators evaluated the technique’s ability to differentiate three important and morphologically similar species of Aspergillus: *A. ochraceus* and *A. westerdijkiae*, and *A. niger*. The mycelium powder methodology they used correctly identified 100% of the set of prediction tests showing that FTIR is also promising for fungal discrimination [43].

Therefore, it can be inferred that this methodology can be used to discriminate different pathogenic agents of interest. Most analyses focus on differentiating species of the same genus or not, based on differences in specific peaks of chemical groups found in pathogens. Thus, FTIR can be applied as a diagnostic method for pathogens of interest in sanitary monitoring, for example, combining robustness, practicality, and reduced time to deliver results.

## 4. Using the FTIR Technique, It Is Possible to Detect Different Types of Molecules in Body Fluids

Applications of FTIR in diagnostics and molecule detection tests can be developed in various body fluids, which can be used either collectively for the same purpose or individually for different analyses. Thus, the construction of a digital signature for each biomolecule creates a detection tool that, combined with the low sample preparation requirements, creates a methodology that can be applied across various species with diverse fluids, such as plasma [44,45], saliva [46,47,48,49], blood [50,51,52], and urine [53,54,55,56], in order to identify a series of marker biomolecules such as proteins and hormones through changes in the spectrum of different small-to-medium-sized molecules that form them, such as acids, peptides, carbohydrates, and lipids, among others [57].

For example, in a pilot study for creating a diagnostic method for myocardial damage due to diabetes [58]. Three different murine samples of the aforementioned fluids were used, indicating changes in lipid patterns, phosphate macromolecules, carbohydrates, and proteins, specifically caused by the disease. In this way, the analysis algorithm perceives the pattern of change in the mentioned bands and, when comparing them with those of the healthy animal, diagnoses it.

SARS-CoV-2 was also the target of FTIR application to detect positive samples in several studies, which showed that photonics can be promising for obtaining simple and rapid diagnostic tests [59]. A group of researchers used samples of a potentially infectious culture supernatant or oral wash from infected mice and saw that controlled UV-inactivated SARS-CoV-2 infection caused strong biochemical changes in the culture supernatant/oral wash. However, SARS-CoV-2 infection induced additional FTIR signals relative to UV-inactivated infection, which correspond to aggregated proteins and RNA. In this way, they managed to establish a simple and robust method for COVID-19 saliva screening [60].

Another interesting example is the diagnosis and classification of susceptible and experimentally immunized groups against respiratory allergies. For this purpose, a study used serum from mice immunized and susceptible to respiratory allergens and managed to separate both groups through detected differences in ester-bound groups, proteins, and collagens [61].

Hormones are also important molecules studied and analyzed by FTIR. These cannot be directly detected through this technique; however, their spectral signatures can be indirectly measured by analyzing the spectral signature of the sample and observing changes in regions associated with wavelength groups functional to the hormone under study [49]. This can be used as an example of stress status, both for humans and animals during physical exertion, through the detection of stress-related hormones such as cortisol and corticosterone [62], as well as the analysis and perception of molecular pattern changes related to neoplastic processes [49].

The diagnosis of such processes can also be observed through the study of other medically important macromolecules. For example, for the diagnosis of digestive tract cancers, a particular research group [51] could not accurately identify which protein was responsible for the condition; however, the diagnosis was made through the standardization of changes in wavelengths related to protein backbones. These differences, because they were repetitive in diseased patients compared to healthy ones, allowed, through an algorithm, the construction of diagnoses based solely on disparities found at that wavenumber [51].

In this context, the same technique was used by another group, to assess the aging of different strains of mice. They observed changes in the secondary structure of hemoglobin proteins, marked by alterations in the spectral signature pattern at the wave number of these structures. In this way, they were able to identify differences related to the aging process characteristic of the different readings of FTIR [63].

Therefore, it can be concluded that this methodology can be used for a series of different body fluids to diagnose multiple contexts. These analyses are performed by comparing the normal state of the spectral fingerprint of the sample with the altered state, where these alterations represent changes in macromolecules, such as hormones and proteins, allowing the construction of diagnoses through the comparison of the sample analysis with that considered standard.

## 5. Would FTIR Applied to Sanitary Monitoring in Laboratory Animals Be an Innovation?

To carry out quality scientific research, it is necessary to control the health of the animals used in experimentation. There are a variety of direct and indirect methods that can be used to detect infectious agents in a population [64]. The use of animals kept in standardized housing conditions has shown great performance in the experimental results of immunopathological research and therapeutic targets, thus reducing the phenotypic variation in results found by different research groups [65]. One of the most important organizations that accredits, and is involved in monitoring the health of experimental animals, is the Federation of European Laboratory Animal Science Associations (FELASA), which recommends monitoring the health of animals such as rodents, cats, dogs, sheep, goats, and non-human primates, among others [64]. For rodents, it is recommended to monitor pathogens such as viruses, bacteria, fungi, and parasites at least every three months; however, it is not a requirement that the animals tested are free of all pathogens for which they were tested [64].

Animal necropsy, for example, allows a systematic examination by inspecting the internal organs and, if any abnormality is found, a sample can be used for histopathological examinations. Histopathology can also be used to detect or confirm infections with non-culturable bacteria such as *Clostridium piliforme*, or extraintestinal parasites such as *Klossiella* spp. [66].Endoparasites, on the other hand, can be diagnosed by PCR [64]. Culture techniques can be used to detect most bacterial and fungal agents. Samples are commonly taken from the genital mucosa, large intestine, nasopharynx, and trachea. Lesions suspected of bacterial origin should be cultured [67].

When considering screening for viruses and some bacteria such as, for example, *Mycoplasma pulmonis*, the use of serological methods is recommended, like bead-based multiplexed fluorescent immunoassay (MFI or MFIA^®^), enzyme-linked immunosorbent assay (ELISA), indirect immunofluorescence assay (IFA), and hemagglutination inhibition assay (HAI), where ELISA and IFA are more sensitive and used as primary tests. Western blots are not suitable for routine screening due to the high cost, but they are highly specific and sensitive, so they are occasionally used for confirmation [64]. Each assay can produce false-positive and false-negative results, so prevalence data can help in estimating the predictive value of test results. Agents considered rarer are less likely to be found in a population and therefore less likely to produce true positives [68,69]. It is noteworthy that due to the complex antigenic structure of bacteria, serological tests applied to detect antibodies in these pathogens present a high risk of false-positive results.

Negative results only mean that antibody activity for the monitored microorganisms was not demonstrated in the animals tracked by the tests used and, therefore, does not necessarily reflect the situation of all animals in the unit. A detectable antibody response can take a few days or weeks for the animals to produce, and therefore serological test results will be negative during the early stages of infection. Studies in rodents infected with some prevalent agents, such as parvovirus, have shown that some infected animals respond poorly and may seroconvert slowly or not at all. Seroconversion also depends on the dose, the biological attributes of the agent, and the genetic makeup, age, and immune status of the infected animal [70,71].

Regarding molecular methods, the use of the PCR (polymerase chain reaction) technique can be useful in confirming positive or doubtful serological results, for the early detection of agents before seroconversion occurs, or for the screening of immunodeficient animals [72]. Many commonly used commercial kits for the identification of pathogenic bacteria may not correctly identify bacterial strains from laboratory animals, such as *Pasteurella pneumotropica* and *Citrobacter rodentium* bacteria. Thus, PCR can be used routinely for the detection and identification of some bacteria not identified in commercial kits, mainly, *Helicobacter* spp., which otherwise are difficult to cultivate or identify at the species level with classical culture techniques [67].

Still, in the molecular context, another important and rapid tool for microbial identification, especially in the case of specific microbiota, is matrix-assisted laser desorption ionization/time-of-flight mass spectrometry (MALDI-TOF MS) [73], the use of which in animal health control exemplifies the importance of spectrometry methods in this context.

The FTIR method has been successfully applied to detect different pathogenic bacteria, thus demonstrating the good quality and efficiency of this technique [55]. Considering the methods described above and the possible delay in obtaining results, the FTIR technique can be a quick and reliable alternative for detecting pathogens for the health control of experimental animals. Furthermore, the technique uses a low number of fluids, thus not requiring animal euthanasia to carry out sanitary control. It is also worth highlighting that techniques aimed at caring for experimental animals reflect on the animal’s well-being, thus improving the results obtained during research (Figure 2).

## 6. In Addition to Diagnosing Infectious Diseases, FTIR also Has the Potential to Be Used in Monitoring the Welfare of Laboratory Animals

Animal welfare is a topic of paramount importance in the science of laboratory animals and in all fields that use them as experimental models, as it can create a responsible, ethical, reproducible, and robust research environment [74,75]. Animal welfare is associated with the way the animal perceives its environment, along with the stimuli present, the sanitary conditions to which it is subjected, and the context that allows the animal to express its natural behavior with minimal discomfort, pain, and stress. These factors are stipulated by the World Organization for Animal Health (WOAH) as the ones that ensure the healthiest animals, as advocated in the five animal freedoms by this agency [76,77,78].

It is important, therefore, that animal models are in a state of welfare as this allows assertiveness in the produced results due to the model’s behavioral and phenotypic normality, ensuring reproducibility of results and absence of bias from behavioral deviations due to stress [79,80]. Thus, those who use animals as models in their experiments should always maintain methodologies to ensure welfare, whether quantitative or qualitative, which will be exemplified in this Section for those applied to murine models.

The most widely used and disseminated scale in murine models is the Mouse Grimace Scale (MGS) [81,82], which exists for other animals with specific adaptations, encoding pain values from 0, absence, to 2, high discomfort. Pain is a parameter that must be controlled and analyzed in many biomedical studies to avoid bias in the acquired results. The absence of pain is directly linked to ethical and responsible work by researchers, except for studies that assess pain [74,83]. In this context, techniques that analyze the animal’s facial expressions compared to what should be considered normal are used as tools to assess the presence and intensity of painful stimuli.

In the search for more assertive and less subjective methods of analyzing animal welfare, methods using implanted telemetric sensors to evaluate the animal’s physiological data, and through the construction of tables interpreted by algorithms with the acquired data, are used to monitor and define the animal’s condition. Despite its high efficiency and fidelity of data, this tool brings many problems that can end up increasing stress in the study condition, arising from the need for moderately invasive surgery, short-term alteration of the animal’s circadian cycle, high cost of the telemetry device, and discomfort caused by the presence of the implant [84].

To reduce potential stress sources arising from the observation and handling of animals described in previous methods, methodologies have been created to remotely, sometimes automatically, to assess animal welfare, both qualitatively and quantitatively [85]. Another option is the use of intelligent racks that use microisolators capable of digitally monitoring animal behavior [86]. These collect behavior and physiology data, such as respiratory rate, and through algorithms, can demonstrate the animal’s state. However, the initial stages of the study and the need for more practical tests, in addition to the need to develop and adapt biomarker analysis algorithms for each case study, hinder this technique [85,86].

Furthermore, there are also camera monitoring methods, which use images in wavelengths ranging from infrared to visible, generating thermal, behavioral, and physiological data that, when interpreted, define the animal’s welfare condition [84,87,88]. It is a highly efficient and minimally invasive method that, by collecting information on vital parameters and behavior, uses computational tools to create highly valid assessments. It is still an expensive method, but with high potential for evolution with the use of deep learning algorithms [89,90].

In murine species, mainly, rats and mice, widely used as experimental models, the main molecule with a stress role is corticosterone [17,91,92,93]. Still, regarding quantitative methodologies, one way to determine animal welfare is related to determining the amount of stress-related hormones in their serum, urine, feces, or other body fluids, mainly glucocorticoids [94]. This analysis can be performed in various ways, with enzymatic immunoassay, radioimmunoassay, liquid chromatography, and mass spectrometry, among others. However, although this tool provides extremely important, highly reliable, and efficient data, its use requires a large number of samples, which often induces stress, especially when blood is used for analysis, due to the constant need for manipulation, discomfort for sample collection, and in some cases, corticosterone measurement depends on animal euthanasia resulting from the withdrawal of a large blood volume.

Therefore, as a response to the problems arising from these commonly used methods, FTIR is a highly efficient tool requiring minimal sample quantities. It is already used in diagnostic tests in humans [62], being non-invasive and reliable. It is known that FTIR is already being used in animal welfare, such as in studies with cows, where molecular and hormonal composition directly in milk is verified to assess the well-being of dairy cattle [95]. In addition, it is used in detecting metabolic changes in fish patterns to assess stress in aquaculture environments [96].

However, the use of this methodology in monitoring these parameters in laboratory animals is not yet widely practiced. Therefore, the use of this tool for monitoring glucocorticoids and other stress-related steroids in murine models is a solution for efficiently and ethically monitoring animal welfare in a research environment [97,98].

## 7. Conclusions

Fourier Transform Infrared Spectroscopy (FTIR) offers a series of significant advantages in identifying molecules in biological samples from humans and animals, such as sensitivity and specificity, speed (a matter of minutes, allowing rapid identification of compounds), having a wide identification range (can identify a wide variety of molecules, including proteins, lipids, carbohydrates, and nucleic acids), being non-destructive (biological samples can be analyzed without being damaged or altered during the process), requiring little sample preparation, and allowing the quantification of molecules (in addition to identification, FTIR can also be used to quantify the concentration of different components), among others. Characteristics that are associated with the importance of diagnostic techniques in animal experimental models are as follows: a small sample volume (more than 80% of the animals used are rodents and small animals, compromising the collection of large sample volumes) and being sensitive and specific, sustainable, and with a low cost, with small sample processing and efficient collection protocols always aiming at animal welfare, which leads us to find that the FTIR technique is very promising in laboratory animals science, both for controlling the health and welfare of animals and for monitoring stress caused by infectious or non-infectious diseases, therefore being an important field of research for the development of technologies that have broad access, contributing to animal health and well-being, the sustainability of research facilities (animal facilities), and the reproducibility of data obtained from animals, reducing the number of animals used in experiments by refining the diagnostic techniques.

## Figures and Tables

**Figure 2 microorganisms-12-00833-f002:**
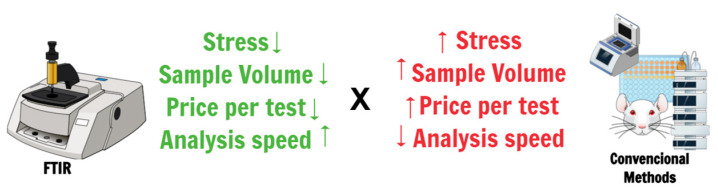
Comparative aspects between the potential that the FTIR technique would have when applied to the sanitary and welfare monitoring of laboratory animals, more specifically, rodents, and the methods currently used (gold-standard). Upward-facing arrows indicate an increase in the indicated parameter, while downward-facing arrows indicate a decrease.

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
