# Peer review of "Perspectives of FTIR as Promising Tool for Pathogen Diagnosis, Sanitary and Welfare Monitoring in Animal Experimentation Models: A Review Based on Pertinent Literature"

_microorganisms, 2024, doi:10.3390/microorganisms12040833_

Round 1

Reviewer 1 Report

Comments and Suggestions for Authors

Dear Authors,

The aim of this review was to collect literature data of the use of the Fourier Transform Infrared Spectroscopy (FTIR) technique in various fields, with particular reference to sanitary welfare monitoring in animal experimentation models.

Given the emergence of this important new rapid diagnostic method, the related studies/review are an added value to the scientific literature for me.

I kindly ask you to consider my comments/suggestions to improve your manuscript and increase your chances for being accepted for publication.

Best regards.

·         Line 24: Replace the word " Furthermore" with " However"

·         Line 87: This isn't quite true. Delete “don’t use reagents”

·         The paragraph 2 “Fourier-Transform Infrared Spectroscopy (FTIR) technique” is too long and detailed. Summary it (E.g. delete from line 116 to 122).

·         On the contrary, paragraph 3 is very little argued despite the fact that there are many paper currently available in the bibliography on the application of FTIR in microbiology. These are just examples. I think they should all be included in a review like this.

o    AlRabiah H, Correa E, Upton M, Goodacre R. High-throughput phenotyping of uropathogenic E. coli isolates with Fourier transform infrared spectroscopy. Analyst. 2013 Mar 7;138(5):1363-9. doi: 10.1039/c3an36517d. PMID: 23325321.

o    Passaris I, Mauder N, Kostrzewa M, Burckhardt I, Zimmermann S, van Sorge NM, Slotved HC, Desmet S, Ceyssens PJ. Validation of Fourier Transform Infrared Spectroscopy for Serotyping of Streptococcus pneumoniae. J Clin Microbiol. 2022 Jul 20;60(7):e0032522. doi: 10.1128/jcm.00325-22. Epub 2022 Jun 14. PMID: 35699436; PMCID: PMC9297836.

o    Manzulli V, Cordovana M, Serrecchia L, Rondinone V, Pace L, Farina D, Cipolletta D, Caruso M, Fraccalvieri R, Difato LM, Tolve F, Vetritto V, Galante D. Application of Fourier Transform Infrared Spectroscopy to Discriminate Two Closely Related Bacterial Species: Bacillus anthracis and Bacillus cereus Sensu Stricto. Microorganisms. 2024 Jan 17;12(1):183. doi: 10.3390/microorganisms12010183. PMID: 38258007; PMCID: PMC10821103.

o    Guerrero-Lozano I, Galán-Sánchez F, Rodríguez-Iglesias M. Fourier transform infrared spectroscopy as a new tool for surveillance in local stewardship antimicrobial program: a retrospective study in a nosocomial Acinetobacter baumannii outbreak. Braz J Microbiol. 2022 Sep;53(3):1349-1353. doi: 10.1007/s42770-022-00774-6. Epub 2022 May 30. PMID: 35644906; PMCID: PMC9433509.

o    Nitrosetein T, Wongwattanakul M, Chonanant C, Leelayuwat C, Charoensri N, Jearanaikoon P, Lulitanond A, Wood BR, Tippayawat P, Heraud P. Attenuated Total Reflection Fourier Transform Infrared Spectroscopy combined with chemometric modelling for the classification of clinically relevant Enterococci. J Appl Microbiol. 2021 Mar;130(3):982-993. doi: 10.1111/jam.14820. Epub 2020 Aug 28. PMID: 32780423.

o    Rebuffo CA, Schmitt J, Wenning M, von Stetten F, Scherer S. Reliable and rapid identification of Listeria monocytogenes and Listeria species by artificial neural network-based Fourier transform infrared spectroscopy. Appl Environ Microbiol. 2006 Feb;72(2):994-1000. doi: 10.1128/AEM.72.2.994-1000.2006. PMID: 16461640; PMCID: PMC1392910.

·         Lines 194-201: Remove or move to paragraph below.

·         Even in paragraph 4 the citations are few compared to what has been done. Also enrich this paragraph e.g. to the list of different body fluids (line 225) add related studies.

·       Blood: Kochan K, Bedolla DE, Perez-Guaita D, Adegoke JA, Chakkumpulakkal Puthan Veettil T, Martin M, Roy S, Pebotuwa S, Heraud P, Wood BR. Infrared Spectroscopy of Blood. Appl Spectrosc. 2021 Jun;75(6):611-646. doi: 10.1177/0003702820985856. Epub 2021 Jan 28. PMID: 33331179.

·       Plasma: Mateus Pereira de Souza N, Hunter Machado B, Koche A, Beatriz Fernandes da Silva Furtado L, Becker D, Antonio Corbellini V, Rieger A. Detection of metabolic syndrome with ATR-FTIR spectroscopy and chemometrics in blood plasma. Spectrochim Acta A Mol Biomol Spectrosc. 2023 Mar 5;288:122135. doi: 10.1016/j.saa.2022.122135. Epub 2022 Nov 20. PMID: 36442341.

·       Saliva: Nascimento MHC, Marcarini WD, Folli GS, da Silva Filho WG, Barbosa LL, Paulo EH, Vassallo PF, Mill JG, Barauna VG, Martin FL, de Castro EVR, Romão W, Filgueiras PR. Noninvasive Diagnostic for COVID-19 from Saliva Biofluid via FTIR Spectroscopy and Multivariate Analysis. Anal Chem. 2022 Feb 8;94(5):2425-2433. doi: 10.1021/acs.analchem.1c04162. Epub 2022 Jan 25. PMID: 35076208.

·       Urine: Sarigul N, Kurultak İ, Uslu GökceoÄŸlu A, Korkmaz F. Urine analysis using FTIR spectroscopy: A study on healthy adults and children. J Biophotonics. 2021 Jul;14(7):e202100009. doi: 10.1002/jbio.202100009. Epub 2021 Apr 7. PMID: 33768707.

·         Line 312: Here, if you want, you could include MALDI-TOF MS as an important and rapid technique in microbial identification.

·         Riga 316: Rimuovere "negli esseri umani".

·         Figura 2: Rimuovere "Eficience" da. Non credo sia corretto affermano che i metodi convenzionali sono meno efficienti dell'FTIR.

Author Response

Dear reviewer,

Firstly, I would like to thank you for your insightful analysis and contributions. Evidently, the final version of the paper is much better. Below I transcribe your considerations and our responses.

 . Line 24: Replace the word " Furthermore" with " However"

      Inserted in the text, word replaced.

  • Line 87: This isn't quite true. Delete “don’t use reagents”

      Inserted in the text, phrase deleted.

  • The paragraph 2 “Fourier-Transform Infrared Spectroscopy (FTIR) technique” is too long and detailed. Summary it (E.g. delete from line 116 to 122).

      Inserted in the text, paragraph summarized.

  • On the contrary, paragraph 3 is very little argued despite the fact that there are many paper currently available in the bibliography on the application of FTIR in microbiology. These are just examples. I think they should all be included in a review like this.

o    AlRabiah H, Correa E, Upton M, Goodacre R. High-throughput phenotyping of uropathogenic E. coli isolates with Fourier transform infrared spectroscopy. Analyst. 2013 Mar 7;138(5):1363-9. doi: 10.1039/c3an36517d. PMID: 23325321.

o    Passaris I, Mauder N, Kostrzewa M, Burckhardt I, Zimmermann S, van Sorge NM, Slotved HC, Desmet S, Ceyssens PJ. Validation of Fourier Transform Infrared Spectroscopy for Serotyping of Streptococcus pneumoniae. J Clin Microbiol. 2022 Jul 20;60(7):e0032522. doi: 10.1128/jcm.00325-22. Epub 2022 Jun 14. PMID: 35699436; PMCID: PMC9297836.

o    Manzulli V, Cordovana M, Serrecchia L, Rondinone V, Pace L, Farina D, Cipolletta D, Caruso M, Fraccalvieri R, Difato LM, Tolve F, Vetritto V, Galante D. Application of Fourier Transform Infrared Spectroscopy to Discriminate Two Closely Related Bacterial Species: Bacillus anthracis and Bacillus cereus Sensu Stricto. Microorganisms. 2024 Jan 17;12(1):183. doi: 10.3390/microorganisms12010183. PMID: 38258007; PMCID: PMC10821103.

o    Guerrero-Lozano I, Galán-Sánchez F, Rodríguez-Iglesias M. Fourier transform infrared spectroscopy as a new tool for surveillance in local stewardship antimicrobial program: a retrospective study in a nosocomial Acinetobacter baumannii outbreak. Braz J Microbiol. 2022 Sep;53(3):1349-1353. doi: 10.1007/s42770-022-00774-6. Epub 2022 May 30. PMID: 35644906; PMCID: PMC9433509.

o    Nitrosetein T, Wongwattanakul M, Chonanant C, Leelayuwat C, Charoensri N, Jearanaikoon P, Lulitanond A, Wood BR, Tippayawat P, Heraud P. Attenuated Total Reflection Fourier Transform Infrared Spectroscopy combined with chemometric modelling for the classification of clinically relevant Enterococci. J Appl Microbiol. 2021 Mar;130(3):982-993. doi: 10.1111/jam.14820. Epub 2020 Aug 28. PMID: 32780423.

o    Rebuffo CA, Schmitt J, Wenning M, von Stetten F, Scherer S. Reliable and rapid identification of Listeria monocytogenes and Listeria species by artificial neural network-based Fourier transform infrared spectroscopy. Appl Environ Microbiol. 2006 Feb;72(2):994-1000. doi: 10.1128/AEM.72.2.994-1000.2006. PMID: 16461640; PMCID: PMC1392910.

Inserted in the text, references were included to support this paragraph.

  • Lines 194-201: Remove or move to paragraph below.

      Inserted in the text, moved to paragraph below, changed to lines 225-232.

  • Even in paragraph 4 the citations are few compared to what has been done. Also enrich this paragraph e.g. to the list of different body fluids (line 225) add related studies.
  • Blood: Kochan K, Bedolla DE, Perez-Guaita D, Adegoke JA, Chakkumpulakkal Puthan Veettil T, Martin M, Roy S, Pebotuwa S, Heraud P, Wood BR. Infrared Spectroscopy of Blood. Appl Spectrosc. 2021 Jun;75(6):611-646. doi: 10.1177/0003702820985856. Epub 2021 Jan 28. PMID: 33331179.
  • Plasma: Mateus Pereira de Souza N, Hunter Machado B, Koche A, Beatriz Fernandes da Silva Furtado L, Becker D, Antonio Corbellini V, Rieger A. Detection of metabolic syndrome with ATR-FTIR spectroscopy and chemometrics in blood plasma. Spectrochim Acta A Mol Biomol Spectrosc. 2023 Mar 5;288:122135. doi: 10.1016/j.saa.2022.122135. Epub 2022 Nov 20. PMID: 36442341.
  • Saliva: Nascimento MHC, Marcarini WD, Folli GS, da Silva Filho WG, Barbosa LL, Paulo EH, Vassallo PF, Mill JG, Barauna VG, Martin FL, de Castro EVR, Romão W, Filgueiras PR. Noninvasive Diagnostic for COVID-19 from Saliva Biofluid via FTIR Spectroscopy and Multivariate Analysis. Anal Chem. 2022 Feb 8;94(5):2425-2433. doi: 10.1021/acs.analchem.1c04162. Epub 2022 Jan 25. PMID: 35076208.
  • Urine: Sarigul N, Kurultak İ, Uslu GökceoÄŸlu A, Korkmaz F. Urine analysis using FTIR spectroscopy: A study on healthy adults and children. J Biophotonics. 2021 Jul;14(7):e202100009. doi: 10.1002/jbio.202100009. Epub 2021 Apr 7. PMID: 33768707.

Inserted in the text, both the exemplified references and others have been added.

  • Line 312: Here, if you want, you could include MALDI-TOF MS as an important and rapid technique in microbial identification.

      Inserted in the text, lines 317-320

  • Riga 316: Rimuovere "negli esseri umani".

      Inserted in the text, phrase removed.

  • Figura 2: Rimuovere "Eficience" da. Non credo sia corretto affermano che i metodi convenzionali sono meno efficienti dell'FTIR.

Inserted in the text, removed from the image.

Reviewer 2 Report

Comments and Suggestions for Authors

According to the title, the manuscript is focused on perspectives and limitations of FTIR as a basic tool efficient for detecting molecules associated with hosts and pathogens in infections, as well as other molecules present in humans and animals’ biological samples. In fact I did not find much information about the limitation of FTIR tool for microorganisms diagnosis, sanitary and welfare monitoring in animal experimentation. That is why I recommend focusing subsection 2. Fourier-Transform Infrared Spectroscopy (FTIR) technique to the proper FTIR (ATR-FTIR) and its limitations for microogranisms diagnosis, sanitary and welfare monitoring.

Author Response

Dear reviewer,

Firstly, I would like to thank you for your insightful analysis and contributions. Evidently, the final version of the paper is much better. Below I transcribe your considerations and our responses.

Row 1 microorganisms not microogranisms

Inserted in the text, word corrected.

According to the title, the manuscript is focused on perspectives and limitations of FTIR as a basic tool efficient for detecting molecules associated with hosts and pathogens in infections, as well as other molecules present in humans and animals’ biological samples. In fact, I did not find much information about the limitation of FTIR tool for microorganisms diagnosis, sanitary and welfare monitoring in animal experimentation. That is why I recommend focusing subsection 2. Fourier-Transform Infrared Spectroscopy (FTIR) technique to the proper FTIR (ATR-FTIR) and its limitations for microogranisms diagnosis, sanitary and welfare monitoring.

 Subsection 3. Infections by different classes of microorganisms can be diagnosed using FTIR on biological samples. Which are different classes of microoorganisms? Staphylococcus aureus, Aspergillus  ochraceus and A. westerdijkiae, and A. niger? Is SARSCoV2 a microorganism? I recommend the authors to pay attention to them.

Inserted in the text. SARSCoV2 paragraph has been changed to section "Using the FTIR technique, it is possible to detect different types of molecules in body fluids." as recommended by the other reviewer.

  Subsection 4. Using the FTIR technique, it is possible to detect different types of molecules in body fluids. I recommend the authors to mention the types of molecules detected with FTIR technique.

Inserted in the text.

  Subsection 5. Would FTIR applied to sanitary monitoring in laboratory animals be an innovation? Authors mention conventional methods for sanitary monitoring animals (infections with Clostridium piliforme, Klossiella spp, Mycoplasma pulmonis etc.). I recommend the authors to prove that FTIR technique has to the same results. 

 Subsection 6. In addition to diagnosing infectious diseases, FTIR also has the potential to be used in monitoring the welfare of laboratory animals. The authors exhaustively present the currently used methods (rows 329-386) and unconvincingly the FTIR technique (rows 387-293). I recommend the authors to provide solid arguments for the FTIR technique.

The literature included provides a basis for the use of FTIR as a promising tool in welfare and health monitoring. Due to its relatively recent use in these fields in work with laboratory animals, there are still no published studies showing data on the specific use of this technology in the context and purpose raised. This response fits in with the comments made by reviewer 2 regarding subsections 5 and 6.

Additionally, given the relevant contributions of this reviewer, which we agree with, we chose to replace the word “microorganisms” in the title with “pathogens”, as it covers all the agents that we have raised in the literature, and which can be discriminated by the FTIR technique, presenting different spectral signatures.

We also changed the mention of the limitations of the technique in the title, because as the reviewers themselves highlighted, we did not focus on the weaknesses, but rather on the solutions that the use of this technique can bring to health control and well-being monitoring.

Round 2

Reviewer 1 Report

Comments and Suggestions for Authors

Dear authors,

the paper was corrected appropriately.

Correct the italicized bacterial names throughout the text.